# Establishment of Bactrian Camel Induced Pluripotent Stem Cells and Prediction of Their Unique Pluripotency Genes

**DOI:** 10.3390/ijms24031917

**Published:** 2023-01-18

**Authors:** Zongshuai Li, Yina Li, Qiran Zhang, Wenbo Ge, Yong Zhang, Xingxu Zhao, Junjie Hu, Ligang Yuan, Wangdong Zhang

**Affiliations:** 1College of Veterinary Medicine, Gansu Agricultural University, Lanzhou 730070, China; 2Gansu Key Laboratory of Animal Generational Physiology and Reproductive Regulation, Gansu Agricultural University, Lanzhou 730070, China; 3College of Life Science and Technology, Gansu Agricultural University, Lanzhou 730070, China; 4Chinese Academy of Agricultural Sciences Lanzhou Institute of Husbandry and Pharmaceutical Sciences, Lanzhou 730070, China

**Keywords:** Bactrian camel, induced pluripotent stem cells, reprogramming, RNA-Seq, pluripotency genes

## Abstract

Induced pluripotent stem cells (iPSCs) can differentiate into all types of cells and can be used in livestock for research on biological development, genetic breeding, and in vitro genetic resource conservation. The Bactrian camel is a large domestic animal that inhabits extreme environments and holds value in the treatment of various diseases and the development of the local economy. Therefore, we transferred four mouse genes (*Oct4*, *Sox2*, *Klf4*, and c-*Myc*) into Bactrian camel fetal fibroblasts (BCFFs) using retroviruses with a large host range to obtain Bactrian camel induced pluripotent stem cells (bciPSCs). They were comprehensively identified based on cell morphology, pluripotency gene and marker expression, chromosome number, transcriptome sequencing, and differentiation potential. The results showed the pluripotency of bciPSCs. However, unlike stem cells of other species, late formation of stem cell clones was observed; moreover, the immunofluorescence of SSEA1, SSEA3, and SSEA4 were positive, and teratoma formation took four months. These findings may be related to the extremely long gestation period and species specificity of Bactrian camels. By mining RNA sequence data, 85 potential unique pluripotent genes of Bactrian camels were predicted, which could be used as candidate genes for the production of bciPSC in the future. Among them, *ASF1B*, *DTL*, *CDCA5*, *PROM1*, *CYTL1*, *NUP210*, *Epha3*, and *SYT13* are more attractive. In conclusion, we generated bciPSCs for the first time and obtained their transcriptome information, expanding the iPSC genetic information database and exploring the applicability of iPSCs in livestock. Our results can provide an experimental basis for Bactrian camel ESC establishment, developmental research, and genetic resource conservation.

## 1. Introduction

The first embryonic stem cells (ESCs) were generated about 40 years ago by culturing mouse inner cell mass (ICM) on feeder layers [1]. ESCs are non-specialized cells and can differentiate into any type [2]. During the next 25 years, these ESCs could only be obtained by separating ICMs from early embryos, but this method is not applicable to large livestock [3,4]. In 2006, researchers first used retroviruses to transfect four mouse genes (*Oct4*, *Sox2*, *Klf4*, and c-*Myc*) into mouse embryonic fibroblasts and obtained cells similar to ESCs, which were termed induced pluripotent stem cells (iPSCs). This technology provides a practical pathway to acquire pluripotent stem cells from large livestock. With the development of this technology, iPSCs have been successfully generated from humans, mice, sheep, goats, cattle, canines, rats, rabbits, pigs, horses, and monkeys [3,4,5,6,7,8,9,10,11,12,13]. With in-depth research on iPSCs, other inducible genes were discovered, including L-*Myc*, N-*Myc*, and *Glis1*, which can replace c-*Myc* [14,15]; *Klf1*, *Klf2*, *Klf5*, *Esrrb*, and *Bmp4* as replacements for *Klf4* [16,17,18]; *Rcor2*, *GMNN*, *TH2A*, *TH2B*, *Obox1*, *Sox1*, and *Sox3* as replacements for *Sox2* [19,20,21,22,23]; whereas *Nr5a1*, *Nr5a2*, *E-cadherin*, *TCL-1A*, *Brn4*, and *Tet1* could efficiently replace *Oct4* [24,25,26,27,28]. However, there are no reports on Bactrian camel induced pluripotent stem cells (bciPSCs) and their unique pluripotency genes.

The Bactrian camel is endemic to desert and semi-desert areas. It possesses a distinctive reproductive physiological structure and can withstand high temperatures, severe cold, and salinity; however, its reproductive efficiency is low. Therefore, it has an important scientific research value. Being a multi-purpose livestock, the Bactrian camel also has important economic value owing to its meat, milk, and wool [29,30]. With research advances, the Bactrian camel has attracted extensive attention in the fields of the treatment of various diseases (such as diabetes and cancer), health care, and immunology [31,32,33,34,35,36]. However, research on the Bactrian camel, particularly its genetic evolution, is still in its preliminary and fundamental stages. Moreover, with modernization, the number of domesticated Bactrian camels is rapidly decreasing, and wild Bactrian camels have become the eighth endangered species due to environmental damage [37]. Therefore, the establishment of Bactrian camel stem cell lines is pertinent to improving and furthering research on the biological development, genetic traits, and genetic breeding of Bactrian camels. Additionally, it will guarantee the in vitro protection of the genetic resources of wild Bactrian camels.

In this study, we used retroviruses to transfer four mouse genes (*Oct4*, *Sox2*, *Klf4*, and c-*Myc*) into BCFFs to obtain bciPSCs, which were identified based on various aspects, including cell morphology, pluripotency markers, differentiation potential, and transcriptome analysis. Moreover, 85 potential pluripotency genes were predicted using the transcriptome analysis of the Bactrian camels. In this study, we aimed to provide an experimental basis for Bactrian camel ESC establishment, developmental research, genetic resource conservation, and basic research and genome data on iPSC enrichment.

## 2. Results

### 2.1. Identification of BCFFs and Plasmid Preparation

We successfully obtained the pMXS-EGFP and the modified pMXS-Sox2 plasmids according to the design scheme (Figure 1(Aa,Ab)) and verified the sequences of the two plasmids via double digestion (Appendix A).

We obtained BCFFs and MEFs (Appendix A) and identified them using specific fibroblast-expressed proteins (Figure 1(Ba,Bb)). *Mycoplasma* was not found in either of the primary cell types (Appendix A). The above results indicate that the two types of cells can be used for the subsequent experiments.

### 2.2. Acquisition and Passage of bciPSCs

In this study, only AP staining was performed before determining the optimal induction culture protocol. The general flow of our experiments is shown in the schematic diagram in Figure 1C. BCFFs were transduced with retroviruses bearing the pluripotency and the reporter genes (Figure 1C), and the cells showed morphological changes after 7 days. The cells were cultured until iPSC clones appeared, which were identified using AP staining (Figure 1Da and Appendix A). After statistical analysis, the best scheme for obtaining and culturing bciPSCs was found to be introducing the four factors of OSKM simultaneously, laying the feeder layer, and adding Lif to E8 as the stem cell culture medium (Figure 1E).

When the colonies grew sufficiently, they were dissociated and transferred to new plates coated with mitomycin C-treated MEFs (Figure 1(Db,Dc)). The cells were passaged when they reached 80% confluency.

### 2.3. Pluripotency Gene Detection

*SOX2* and *NANOG* expression was detected in seven bciPSC cell lines (Figure 1F)

### 2.4. Telomerase Gene Expression in bciPSCs

Telomerase (*TERT*) gene expression was significantly increased in all three bciPSC lines (Figure 1G).

### 2.5. Karyotype Analysis

The karyotypes of bciPSC-A2 in P12, bciPSC-A3 in P15, and bciPSC-B13 in P10 were normal diploids (2n = 74) [38] (Figure 2A).

### 2.6. Immunocytochemistry

In the 11th-generation bciPSC-A3 cells, nine pluripotency-related proteins were selected for immunofluorescence detection, and all marker proteins were expressed (Figure 2B).

### 2.7. EB Formation

We inoculated the three lines of bciPSCs (Figure 3A) into Petri dishes and cultured them in differentiation medium for 7 days to obtain EBs (Figure 3A and Appendix A). We then detected germ layer marker genes by PCR. We found that endoderm (*FOXA2*), mesoderm (*MYOD1*, *DES*, and *MYF5*), and ectoderm (*GFAP*, and *PAX6*) genes were expressed in the differentiated cells but not in undifferentiated cells (Figure 3B). After differentiation and additional culture, immunofluorescence analysis revealed that EBs were positive for FOXA2 (endoderm), DES (mesoderm), GFAP (ectoderm), and β-tubulin (reference gene) (Figure 3C). The results show that bciPSCs can differentiate into three-germ layer cells in vitro.

### 2.8. Teratoma Formation

Following bciPSC injection into nude mice, only the bciPSC-A3 cell line formed solid tumors after 4 months, whose sizes were approximately 5 × 3 × 3 mm^3^ (Appendix A). HE staining revealed that these tumors were neoplastic (Figure 3(Ea)) and contained a variety of tissue cells, including nerve cells (ectoderm) (Figure 3(Eb)), connective tissue (mesoderm) (Figure 3(Ec)), and glands (endoderm) (Figure 3(Ed)). Immunohistochemical results showed that the three germ layer proteins, GFAP (ectoderm) (Figure 3(Da), Db positive; Dc negative), DES (mesoderm) (Figure 3(Dd), De positive; Df negative), and FOXA2 (endoderm) (Figure 3(Dg), Dh positive; Di negative), in the teratoma were positive with abundant cell morphology. After staining the mesoderm for collagen (Figure 3(Fa,Fb)) and reticular fibers (Figure 3(Fc,Fd)), we found abundant knot connective tissue.

### 2.9. RNA-Seq analysis

After receiving transcriptome data, 12 randomly selected genes were tested by qRT-PCR, and their log values were plotted. The results showed consistent trends between the qRT-PCR and the transcriptome, indicating that the sequencing results are accurate and reliable (Appendix A).

The correlation heat map and sample clustering diagram revealed high correlation and close affinity between the seven bciPSC lines (Appendix A). The violin chart demonstrated that the sequencing data had only a few abnormal values and was highly reliable (Appendix A). The upregulated and downregulated genes in each transcriptome dataset were mapped (Figure 4A). The results indicated that the expression of a large number of genes changed during the reprogramming of BCFFs and that the reprogramming direction was similar.

We selected 7561 significantly (|log_2_FC| ≥ 1, *p* < 0.05) upregulated and downregulated genes from seven bciPSC lines to create a heatmap (Figure 4B). To create a non-clustering heatmap, these differentially expressed genes were mapped to the microarray data of humans, mice, and pigs (Figure 4B). A Venn diagram was constructed using the significantly differentially expressed genes of the four species (Figure 4C). The results indicated that bciPSCs had 2056 (human), 408 (mouse), and 1317 (pig) genes in common, indicating that bciPSCs were more similar to human ESC and pig iPSCs. Therefore, during the follow-up analysis, we focused mainly on humans and pigs as supplemental datasets.

The top 20 GO term bubble diagrams of bciPSCs indicated that they mainly belong to cellular components (Appendix A). The enrichment circle maps of the signal pathway with a q value < 0.05 indicated that the pathway related to genetic information processing was activated (Appendix A). Next, we selected the signal pathways related to stem cell formation (apoptosis, oxidative phosphorylation, signaling pathways regulating pluripotency of stem cells, and PI3K, MAPK, Wnt, JAK-STAT, and TGF-β signal pathways [39,40,41,42,43,44,45]) in addition to the signal pathways with a q value < 0.05, and assembled them into a network diagram. The results showed that MAPK and cell cycle signal pathways were at the core (Appendix A). In the aforesaid studies, seven bciPSCs were associated with 52 signaling pathways, from which six studied pathways were selected (cell cycle, p53, apoptosis, cellular senescence, cancer, DNA replication, and oxidative phosphorylation signaling pathways [46,47]) to draw a heat map (Figure 4D). The heat maps of DNA replication, apoptosis, and cell cycle signaling pathways [46] have been displayed separately (Appendix A). Overall, bciPSC and human ESC had remarkably similar gene expression patterns for these pathways.

During reprogramming, alterations in *SAP30* and *CDKN1A* (*P21*) were consistent with those of mouse and human stem cells (Appendix A) [48,49,50]. Moreover, *Sox2* and *NANOG,* detected by qRT-PCR, were not effectively detectable in RNA-Seq data, while *Klf4* and c-*Myc* genes were down-regulated in bciPSCs (Figure 4E). This situation also occurred during the establishment of pig iPSCs, possibly because the reference genome, quality control of transcriptome data, or gene expression did not reach the detection threshold [6]. However, we found the expression of the replacement genes increased, such as *GRB2* for *Oct4* [51]; *RCOR2* [20], and *GMNN* [51] for *Sox2*; L-*MYC* for c-*Myc* [15], and L-Myc’s binding protein gene *MYCBP* (Figure 4E). Additionally, the pluripotency-related genes *GDF3*, *NFE2L3*, *GABRB2*, *PODXL*, *PODXL2*, *GNL3*, *DNMT3B*, and *TERT* were overexpressed in bciPSCs (Figure 4E). Furthermore, these gene expression trends were consistent with those in human ESCs and porcine iPSCs [6,52], including *PDLIM1*, *PTPN12*, *NFIB*, *BMP4*, *NEFL*, *LTA4H*, and *TCEA3* (Figure 4E).

We also created a heat map of DNA methyltransferases (*DNMT1*, *DNMT3A*, and *DNMT3B*), DNA methyl oxidases (*TET1* and *TET3*), and histone demethylases (*KDM4B*, *KDM2A*, *KDM1A*, and *KDM3A*) expressed in bciPSCs. The changes in their expression were consistent with reported iPSC formation and similar to those in human ESC (Figure 4F) [53,54,55,56]. We also found that histones were overexpressed in the bciPSC sequencing data, similarly as in human ESC (Figure 4G) [57].

### 2.10. Preliminary Mining of Potentially Pluripotency Genes of bciPSCs

In the direct analysis of sequencing data by screening the top 20 GO terms (Appendix A) and 52 signal pathways implicated in the pluripotency verification (*2.10. RNA-Seq analysis*) (Appendix A), 511 genes were assigned to group 1.

Based on the analysis of the reported inducible genes, 43 inducible genes were identified (Appendix A), and 34 were detected in the sequencing data of bciPSCs. The distribution of 34 genes in the GO and KEGG databases is displayed in an alluvial plot (Figure 5Aa, Ab) to observe their distribution more intuitively. The primary enriched pathways included KEGG-A-class (cellular processes), KEGG-B-class (cellular community–eukaryotes), and pathway ko04550 (Appendix A). The main enriched GO terms were GO components (GO:0043231, GO:0043232, and GO:0044451), GO functions (GO:0000982, GO:0000977, GO:0003712, GO:0046983, GO:0000976, and GO:0005515) and GO processes (GO:0001101, GO:0006357, GO:0019827, and GO:0008284) (Appendix A). After summarization, 2934 upregulated genes were obtained, of which 401 belonged to group 2.

The venn diagram of the two data groups (Figure 5B) revealed an insignificant difference between the groups, which therefore could be combined for analysis. According to the analysis of five network diagrams using the STRING database with a total number of proteins ≥6 (Figure 5C), 46 proteins had over three protein connections (Figure 5D). Eventually, 46 core genes were obtained. The 42 complementary genes were obtained by summarizing the first 30 genes of each bciPSC line (Figure 5D). After comparing the core and complementary genes, a total of 85 genes were obtained, three of which were shared by both (Figure 5D).

## 3. Discussion

Although the production of and research on iPSCs is growing, it remains limited to human and model organisms. Establishing iPSC lines and genetic information databases for various non-model organisms is significant for studying iPSC mechanisms and the resource development of non-model organisms. Based on previous reports on the establishment of iPSCs in other species [3,4,5,6,7,8,9,10], we selected the retroviral system with the broadest host range to import inducible genes. The comparison of different culturing methods revealed that trophoblasts and Lif can promote the generation of bciPSCs and maintain their stable passage [42,43]. In an optimal cultural environment, the reprogramming efficiency is approximately 0.036%. Compared with that of iPSCs of other species, the induction system of bciPSCs needs to be improved [6]. Some cells undergo morphological alterations on the seventh day after exogenous gene insertion; however, stem cell clones arise later, which may be species-specific.

We verified the pluripotency of bciPSCs using various methods. bciPSCs have a high nucleocytoplasmic ratio, show epithelioid cell morphology, express pluripotency genes, are AP-positive, and have normal diploid chromosomes (2n = 74) [38]. Moreover, *TERT*, a marker of ES cells in several species, was active [6,58,59]. The immunofluorescence results of SSEA1, SSEA3, SSEA4, TRA-1-60, and TRA-1-81 were positive, which were different from the results of other species, perhaps due to the difference in reprogramming in different species [4,60]. bciPSCs can differentiate into the three germ layers in vitro and in vivo, and only bciPSC-A3 can form teratomas, possibly related to the test procedure or differentiation ability [6,60,61]. The teratoma formation time was 4 months or even longer, and this phenomenon has also been observed during the identification of horse iPSCs [7], which may be related to the extremely long gestation period of the Bactrian camels.

The RNA-Seq results showed that bciPSCs and BCFFs had many distinct genes, and the overall modifications were similar to those of human ESC. We further analyzed the pluripotency-related signaling pathway and pluripotency-related genes and DNA methylases, and the results indicated that the bciPSCs are pluripotent. We also found significant differences among the stem cells of Bactrian camels, humans [52], pigs [6], mice [62], bovines [63], and other species, suggesting that the Bactrian camel may have unique pluripotency-related genes. Therefore, establishing iPSCs in non-model organisms is crucial for studying the induction mechanism of iPSCs.

According to the screening of reported inducible genes, 34 were primarily distributed in intracellular membrane-bound organelles and regulated transcription from the RNA polymerase II promoter of cell processes in the GO database. The KEGG database had four signalaling pathways: signaling pathways regulating the privacy of stem cells, pathways in cancer, the Hippo signaling path, and the TGF-beta signaling path. Four hundred and one genes were obtained after screening. According to the direct analysis of RNA-Seq data, 511 genes were obtained. After a comparison of the two data sets, 46 core genes were obtained. We also selected 42 supplementary genes according to the order of differential gene expression levels. Comparing core and supplemental genes revealed very few shared genes. This shows that supplementary genes can also compensate for subjective screening. Through RNA-Seq data mining of bciPSCs, we predicted 85 potentially unique pluripotency genes in Bactrian camels, which can be used as potential marker genes for bciPSCs. They are mainly related to cell proliferation (such as *CYTL1*, *TTK*, *CDC45*, *ASF1B*, *SGO1*, *CDC6*, *CENPW*, *KIAA0101*, *AREG*, *DTL*, *HJURP*, and *SAPCD2*) [64,65,66,67,68,69,70,71,72,73,74], cancer cell infiltration (such as *MEOX2*, *CTSC*, and *CSRP2*) [75,76,77], production and maintenance of pluripotent stem cells (such as *PROM1*, *FOXM1*, *CDCA5*, *CD74*, *MEST*, *FZD*, and *RAD54L*) [78,79,80,81,82,83,84], and promotion of cancer (such as *CLSTN3*, *DLG2*, and *E2F8*) [84,85,86]. Based on the gene regulation and expression levels of 85 genes, we further speculate that *ASF1B* [68], *DTL* [73], *CDCA5* [79], *PROM1* [82], *CYTL1* [71], *NUP210* [87], *Epha3* [88], and *SYT13* [89] are particularly important. Our results support studies on the identification of Bactrian camel-specific inducible genes, optimization of the induction system, and the study of the iPSC induction mechanism. However, further verification will be required.

## 4. Materials and Methods

### 4.1. Plasmid Preparation

The mouse inducible pluripotency factor-carrying plasmids pMXS-Oct4, pMXS-Sox2, pMXS-Klf4, and pMXS-cMyc and the retroviral packaging plasmids pCMV-VSV-G and pUMVC were stored in our laboratory. Using *EGFP* as a template, we designed primers (Appendix A) that were cloned and ligated into the HindIII/BamHI site of the pMXS plasmid to generate the pMXS-EGFP reporter plasmid. We introduced a pair of bases (TA) into the original pMXS-Sox2 plasmid (Appendix A), complemented the three missing amino acids, and constructed a redesigned pMXS-Sox2 plasmid for our investigations. All plasmids were re-sequenced to evaluate their integrity before use.

### 4.2. Cell Lines and Cell Culture Conditions

BCFFs were derived from 3-month-old Bactrian camel fetuses and grown in a tissue block adherent culture after digestion and separation [90]. The cells were maintained at 37 °C and 5% CO_2_ in DMEM/F12 glucose (Hyclone, Logan, UT, USA) supplemented with 10% fetal bovine serum (FBS) (BI, Kibbutz Beit Haemek, Israel). The cells were subcultured once they reached 90% confluence. The first three generations of BCFFs were preserved in liquid nitrogen.

Mouse embryonic fibroblasts (MEFs) were derived from 13.5-day-old mouse embryos. The culture method and number of cryopreserved generations were the same as those of BCFFs. Mitomycin C (Sigma, St. Louis, MO, USA)-treated MEFs were coated on plates to increase the bciPSC number.

The two types of fibroblasts obtained in primary culture were identified as follows. The cells were fixed in 4% paraformaldehyde for 30 min at 25 °C. After washing twice with phosphate-buffered saline (PBS) (BI, Kibbutz Beit Haemek, Israel), the cells were treated with PBS containing 0.2% Triton X-100 for 20 min. The cells were then incubated in PBS supplemented with 5% bovine serum albumin (BSA) (Sigma, St. Louis, MO, USA) for 30 min. After three washes with PBS, the cells were incubated overnight at 4 °C in PBS containing the primary antibody anti-mouse Vim (1:200; Santa, Dallas, TX, USA; sc-6260). The secondary antibody used was Alexa488-conjugated goat anti-mouse IgG (1:400; Abcam, Cambridge, UK; ab150113). The cells were imaged after incubation with the secondary antibody diluted in PBS for 1 h at 37 °C in the dark. The cells used in the experiment were assessed for *Mycoplasma* contamination using the *Mycoplasma* Staining Detection Kit (Beyotime, Shanghai, China).

The Lanzhou Veterinary Research Institute of the Chinese Academy of Sciences donated 293T cells. The cells were cultured in DMEM/high glucose (Hyclone, Logan, UT, USA) supplemented with 10% FBS and incubated at 37 °C under 5% CO_2_.

All experimental procedures were approved by the Animal Care and Use Committee of the College of Veterinary Medicine at Gansu Agricultural University.

### 4.3. bciPSC Culture Method Analysis

We used 293T cells to package five retroviruses, four of which contained inducible pluripotency genes (*Oct4*, *Sox2*, *Klf4*, and *c-Myc*), and one contained a reporter gene (*EGFP*), and transduced them into laboratory-preserved BCFFs. The culture medium was replaced with an iPSC induction medium composed of KnockOut DMEM/F12 medium (Gibco, Grand Island, NY, USA) supplemented with 20% KnockOut Serum Replacement (Gibco, Grand Island, NY, USA), 2 mmol non-essential amino acids (NEAAs) (Gibco, Grand Island, NY, USA), 0.1 mmol β-mercaptoethanol (Sigma, St. Louis, MO, USA), and 1 mM valproic acid (Sigma, St. Louis, MO, USA). The cells were subcultured after seven days. However, because of the lack of certainty in producing these cells, we evaluated six different subsequent culture protocols (Appendix A) by assessing the number of ESC-like colonies obtained from retrovirus-transduced BCFFs on day 34. Based on the results, we selected the optimal induction culture protocol, which was used for all future experiments.

### 4.4. bciPSC Generation

We produced bciPSCs according to the previously obtained optimal culture protocol. On day 34, the clones of bciPSCs were selected using a mechanical method for subculture. The obtained bciPSCs were divided into two groups: group A (originally preserved in the laboratory) and group B (supplemented with a missing base pair) according to the different pMXS-Sox2 plasmids in the induction combination.

### 4.5. Pluripotency Gene Detection

The detection primers for pluripotency genes were designed using the NCBI Primer-BLAST tool (Appendix A), and the transcripts of bciPSC lines were analyzed using quantitative real-time polymerase chain reaction (qRT-PCR). Based on the results, BCFFs and seven bciPSC lines were used for transcriptome sequencing. bciPSC-A2, bciPSC-A3, and bciPSC-B13 were randomly selected from seven bciPSC lines for the subsequent culture and verification.

### 4.6. qRT-PCR

Total RNA was extracted from cells using TRIzol reagent (Sigma, St. Louis, MO, USA) according to the manufacturer’s protocol and stored at −80 °C. cDNA was prepared using the RevertAid First Strand cDNA Synthesis Kit (Thermo Fisher Scientific, Waltham, MA, USA) following the manufacturer’s protocol. Quanti Nova SYBR Green PCR Kit (Qiagen, Dusseldorf, Germany) was used for qRT-PCR assays. All qRT-PCR reactions were performed on a Veriti 96-well thermal cycler (Bio-Rad, Hercules, CA, USA). This method was used to detect endogenous genes (such as n*SOX2* and n*NANOG*) (Appendix A), the telomerase gene (*TERT*) (Appendix A), and RNA-Seq results (Appendix A). Specific primers were designed using the NCBI Primer-BLAST tool to amplify fragments corresponding to the selected genes. All samples were amplified in triplicate, and the mean and standard error values were calculated. Expression levels of all genes were calculated using the 2^−ΔΔCT^ method and were normalized to β-actin.

### 4.7. Alkaline Phosphatase Staining

Alkaline phosphatase (AP) staining was performed using the Alkaline Phosphatase Detection Kit (Beyotime, Shanghai, China).

### 4.8. Karyotype Analysis

The cells were transferred to a T25 bottle. When their density reached approximately 70%, colchicine (Sigma, St. Louis, MO, USA) was added to the final concentration of 0.25 µg/mL; the cells were digested and collected with trypsin after a 2.5 h treatment. The cells were then resuspended in 8 mL of potassium chloride solution (37 °C; 0.075 M), allowed to stand at 37 °C for 30 min, and 1 mL of freshly prepared fixative (glacial acetic acid: methanol = 1:3) was added to collect the cells. A few drops of the fixative were slowly added and mixed well, and another 8 mL was added to collect the cells. The cells were resuspended in 0.3 mL of the fixative, and 2–3 drops of the suspension were dropped onto the 4 °C glass slide. The slide was blown onto and baked several times with an alcohol lamp. The slide was then treated with 1% Giemsa solution (Beyotime, Shanghai, China) dye solution for 10–15 min; the seal film was washed off, observed, and micrographed with an oil lens. As there was no standard banding mode map for Bactrian camels, only a statistical analysis of the number of chromosomes was performed. The number of three bciPSCs for mitotic-phase spreads assessment was 22 (bciPSCs-A2), 25 (bciPSCs-A3), and 20 (bciPSCs-B13), respectively.

### 4.9. Immunofluorescence

Immunofluorescence was performed using the same steps as in fibroblast identification, outlined above. Primary antibodies included anti-mouseOCT4 (1:200; Santa, Dallas, TX, USA; sc-5279), anti-mouseSOX2 (1:200; Santa, Dallas, TX, USA; sc-365823), anti-mouseNANOG (1:200; Santa, Dallas, TX, USA; sc-374103), anti-mouseTRA-1-60 (1:200; Santa, Dallas, TX, USA; sc-21705), anti-mouseTRA-1-80 (1:100; Santa, Dallas, TX, USA; sc-21706), anti-mouseSSEA1 (1:200; Santa, Dallas, TX, USA; sc-21702), anti-mouseSSEA3 (1:200; Santa, Dallas, TX, USA; sc-21703), anti-mouseSSEA4 (1:200; Santa, Dallas, TX, USA; sc-21704), and anti-mouseREX1 (1:200; Santa, Dallas, TX, USA; sc-377095). The secondary antibody used was Alexa647-conjugated goat anti-mouse IgG (1:400; Abcam, Cambridge, UK; ab150115).

### 4.10. In Vitro Differentiation

Three bciPSC cell lines were harvested by treatment with trypsin (0.25%, BI, Kibbutz Beit Haemek, Israel) and transferred to 35 mm Petri dishes (Corning, Corning, NY, USA) in DMEM/F12 containing 10% FBS (BI, Kibbutz Beit Haemek, Israel), 2 mM L-glutamine (Sigma, St. Louis, MO, USA), 1% NEAA (Sigma, St. Louis, MO, USA), and 0.1 mM β-mercaptoethanol (Sigma, St. Louis, MO, USA) to induce embryonic body (EB) formation. After seven days of suspension culture, EBs were transferred onto gelatin-coated tissue culture plates and cultured in a medium supplemented with 10% FBS for another seven days. The three germ layer marker genes were detected using qRT-PCR, and protein levels were identified by immunofluorescence. Primary antibodies include anti-rabbit FOX2 (1:400; CST, Danvers, MA, USA; 8186T), anti-rabbit DES (1:400; CST, Danvers, MA, USA; 5332T), anti-mouse GFAP (1:300; CST, Danvers, MA, USA; 3670T), and anti-mouse β-tubulin (1:200; Santa, Dallas, TX, USA; sc-5274). The secondary antibodies were Alexa647-conjugated goat anti-mouse IgG (1:400; Abcam, Cambridge, UK; ab150115) and Alexa647-conjugated goat anti-rabbit IgG (1:400; Abcam, Cambridge, UK; ab150083).

### 4.11. Teratoma Formation

To evaluate pluripotency in vivo, we used matrix glue (Thermo Fisher Scientific, Waltham, MA, USA) to attach BCFFs and three bciPSC lines (1 × 10^7^ cells/site), which were then resuspended and transplanted into the dorsal flank of immune-deficient BALB/cNude mice. Three mice were used for each cell, for a total of twelve mice. Six- to eight-week-old nude mice were housed under pathogen-free conditions in a temperature-controlled room on a 12/12 h light/dark schedule with food and water *ad libitum*. After four months, the mice were sacrificed, and the tumors were dissected. Teratomas were then analyzed by hematoxylin and eosin (HE) staining (Solarbio, Beijing, China), a reticular fiber staining kit (Solarbio, Beijing, China), and collagen fiber staining (Solarbio, Beijing, China). Simultaneously, the three-germ layer identification antibody was used for the immunohistochemical identification of teratomas.

### 4.12. RNA-Seq

Because Bactrian camels do not have corresponding gene chips for microarray analysis, we used transcriptome sequencing technology. Eight samples (BCFFs, bciPSC-A1, bciPSC-A2, bciPSC-A3, bciPSC-A6, bciPSC-A7, bciPSC-A23, and bciPSC-B13) were selected for transcriptome sequencing. The cell samples were sent to Genedenovo Biotechnology Co., Ltd. (Genedenovo, Guangzhou, China) on dry ice to establish a cDNA library and for subsequent analyses.

HISAT2 was used to align RNA sequencing reads, and the Camelus bactrianus Ca_bactrianus_MBC_1.0 genome assembly (https://www.ncbi.nlm.nih.gov/genome/?term=Camelus+bactrianus (accessed on 24 September 2021)) was used as the reference genome. RNA-seq and microarray data were analyzed using edgeR and GEO2R software, respectively.

### 4.13. Heatmap Generation and Data Visualization

Since there is a dearth of information and research on Bactrian camel stem cells, we retrieved databases of microarray data from human [52], mouse [62], and pig [6] stem cells microarray data from Gene Expression Omnibus (GEO) DataSets (Human, GSE33298; Mouse, GSE51483; Pig, GSE15472) as references and used these for subsequent data analysis. We screened the microarray data of these three species according to the conditions of |log_2_FC| ≥ 1, *p* < 0.05.

The RNA-Seq sequencing results were verified using qRT-PCR and analyzed. We summarized the signal pathways with q < 0.05 and the first 20 items of Gene Ontology (GO) analysis used to observe the main changes in the genetic information. We compared the differentially expressed genes in the RNA-Seq data and the GEO dataset. We also analyzed the enriched signaling pathways, pluripotency-related genes, and important genes in the reprogramming process in RNA-Seq data. The online software of the sequencing company was used to visualize the sequencing data.

### 4.14. Preliminary Mining of Potentially Inducible Genes in bciPSCs

First, we used RNA-Seq sequencing data from bciPSCs to perform direct screening. We summarized upregulated genes involved in the signaling pathways identified during data visualization and the first 20 terms of the GO analysis in the seven bciPSCs. The upregulated genes were screened using counts ≤50 and ≥20 in BCFFs and bciPSCs, respectively, and labeled as group 1. Next, we performed screening based on known induced genes. Based on the relevant literature review, we compiled a comprehensive list of inducible genes across different species. Each bciPSC gene in the RNA-Seq data was annotated using the Kyoto Encyclopedia of Genes and Genomes (KEGG) database (KEGG-A, KEGG-B, and pathways) and the GO databases (GO components, GO functions, and GO processes). Subsequently, we selected the main subdatasets containing the summarized inducible genes at the six annotation levels and screened the upregulated genes. The upregulated genes in the new data set were screened using counts of ≤50 in BCFFs and ≥20 in bciPSCs and designated as group 2.

Ven map analysis for groups 1 and 2 determined whether to combine the analyses based on the degree of dissimilarity between the gene sets. After confirmation, the screened genes were imported to the Search Tool for the Retrieval of Interacting Genes database(STRING, http://string.embl.de/ (accessed on 28 November 2021)), to both predict gene interactions and select core genes. The Bactrian camels were not included in the reference species list. Accordingly, the most closely related *Bos taurus* was selected as the reference species.

To compensate for the disadvantages of selecting groups 1 and 2, we screened the upregulated genes without any screening treatment using counts of ≤50 in BCFFs and ≥20 in bciPSCs and designated the gene set as group 3. The genes of each of the seven bciPSCs in group 3 were sorted in descending order of copy number. The first 30 genes were summarized and used as supplementary genes. Finally, the core genes and complementary genes were combined to form the final candidate gene set.

## 5. Conclusions

This is the first study to obtain bciPSCs with stable cell morphology, expression of pluripotency genes and markers, and the ability to differentiate in vitro and in vivo. Furthermore, we predicted 85 candidate genes that can be used for reprogramming Bactrian camel cells. Our efforts in transcriptome data mining and basic research support the development of the iPSC genetic information database and the use of iPSC in livestock by providing biological materials and data for establishing the Bactrian camel ESC, developmental research, genetic breeding, and the conservation of genetic resources. Considering the development of research on the disease treatment of Bactrian camels, this research is expected to contribute to the transformation of regenerative medicine in the future.

## Figures and Tables

**Figure 1 ijms-24-01917-f001:**
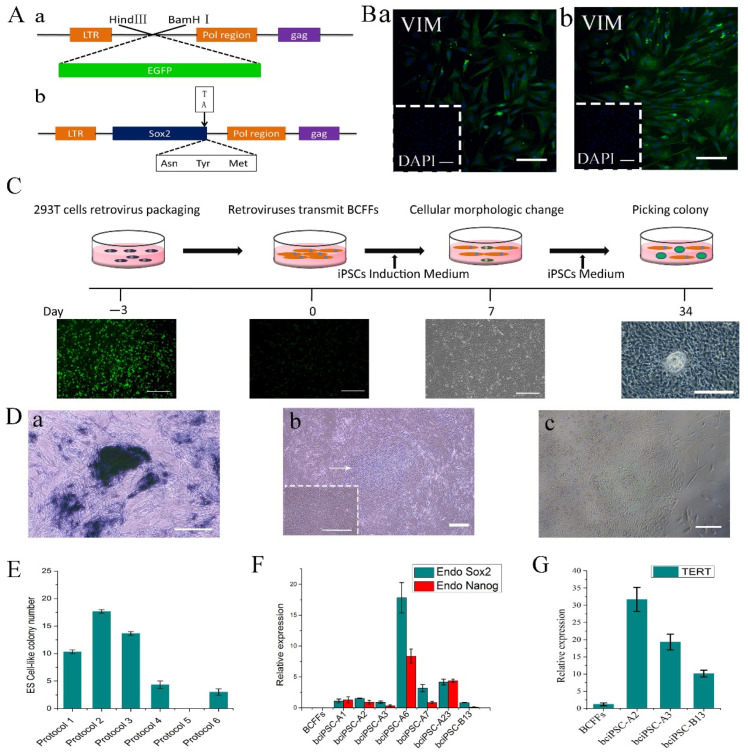
Acquisition of bciPSCs. (**A**) **a**, PMXS-EGFP plasmid map; b, modified PMXS-Sox2 plasmid map; (**B**) **a**, vim protein immunofluorescence identification of BCFFs (20×, 50 μm), **b**, vim protein immunofluorescence identification of MEFs (20×, 50 μm); (**C**) schematic diagram of the process; (**D**) **a**, AP positive clone (20×, 50 μm), **b**, bciPSCs clone (4×, 250 μm; 20×, 50 μm), **c**, P1 of bciPSCs (10×, 100 μm); (**E**) histogram of the number of AP positive clones obtained by six cell induction culture protocols; (**F**) qRT-PCR results of *Sox2* and *Nanog* in BCFFs and seven bciPSC lines; (**G**) telomerase activity detection of bciPSCs and BCFFs.

**Figure 2 ijms-24-01917-f002:**
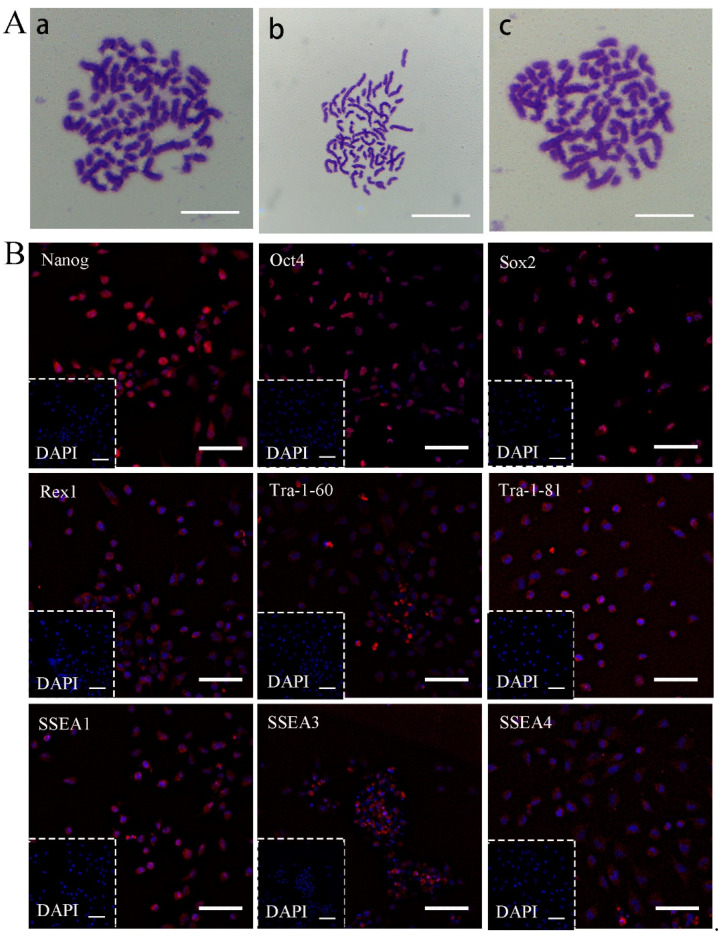
The epigenetic features and pluripotency of bciPSCs. (**A**) The karyotype of **a**, bciPSC-A2, **b**, bciPSC-A3, and **c**, bciPSC-B13 were 74 (100×, 10 μm); (**B**) Immunofluorescence identification of P11 bciPSC-A3. Nanog, Sox2, Oct4, SSEA1, SSEA3, SSEA4, Rex1, Tra-1-60, and Tra-1-81 were all positive (20×, 50 μm).

**Figure 3 ijms-24-01917-f003:**
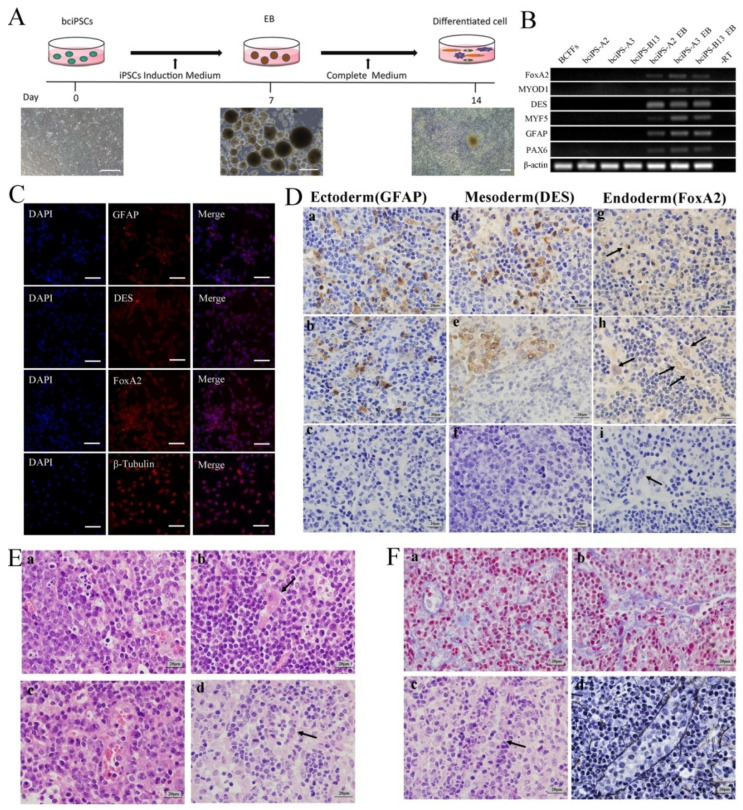
Embryoid body differentiation and teratoma formation. (**A**) flow chart of embryoid body formation and differentiation of bciPSCs; (**B**) PCR detection of bciPSCs and EB cells Germ layer gene expression; (**C**) immunofluorescence detection of three germ layer marker genes in EB cells, FoxA2 (endoderm), DES (mesoderm), GFAP (ectoderm) and β-Tubulin (reference gene) (20×, 50 μm); (**D**) immunohistochemical results of teratoma (100×, 20 μm), (**a**,**b**) GFAP (ectoderm). (**d**,**e**) DES (mesoderm), (**g**,**h**) FoxA2 (endoderm), (**c**,**f**,**i**) as the corresponding negative control; (**E**) HE staining results (100×, 20 μm), (**a**) proved to be the mitotic phase of the tumor, (**b**) nerve cells (ectoderm), (**c**) connective tissue (mesoderm), (**d**) glands (endoderm); (**F**) special histochemical staining for teratoma (100×, 20 μm), (**a**,**b**), collagen fiber staining (blue), (**c**) reticular fibers under HE staining, (**d**) reticular fibers stained (black).

**Figure 4 ijms-24-01917-f004:**
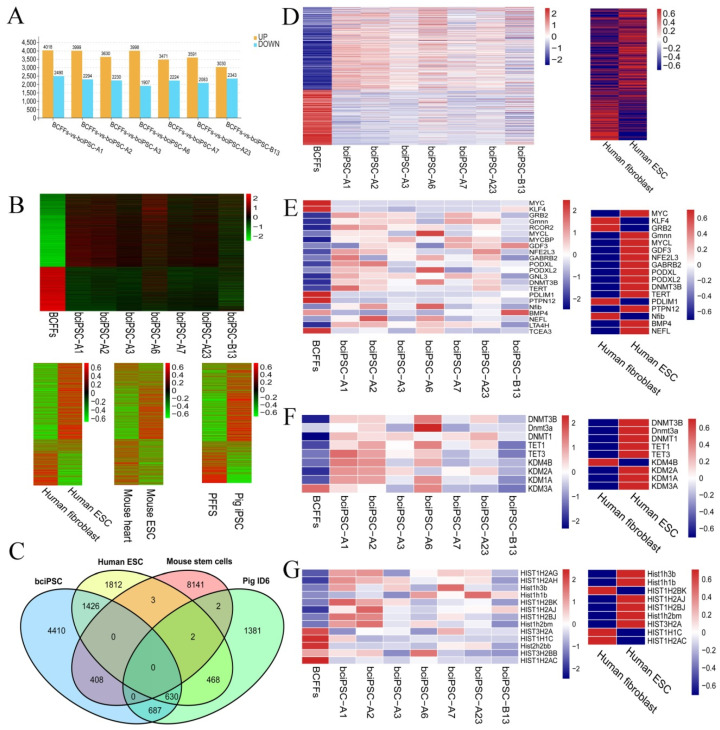
RNA-Seq verifies the pluripotency of bciPSCs. (**A**) differential gene histogram, (**B**) non-clustering heat maps of significantly up-regulated and down-regulated genes in bciPSC lines, and non-clustering heat map of significantly different genes in Bactrian camel sequencing data mapped to mouse ESC cell, human ESC cell, and pig iPSC microarray data, (**C**) venn diagrams of the significantly different genes in the four species of ESC and iPSC. The screening conditions for significantly different genes were |log_2_FC| ≥ 1, FDR < 0.05, (**D**) heatmaps made of genes in all signaling pathways of interest in bciPSCs, and the results of the mapping of corresponding genes into human ESC, (**E**) heatmaps of pluripotency genes and pluripotency-related gene expression in Bactrian camel samples, and heatmaps when each gene was mapped to human ESC, (**F**) heatmap of DNA methyltransferases, DNA methyl oxidases, and histone demethylases in bciPSC and human ESC, (**G**) heatmap of all histone genes in bciPSC sequencing data, and the results mapped into human ESC.

**Figure 5 ijms-24-01917-f005:**
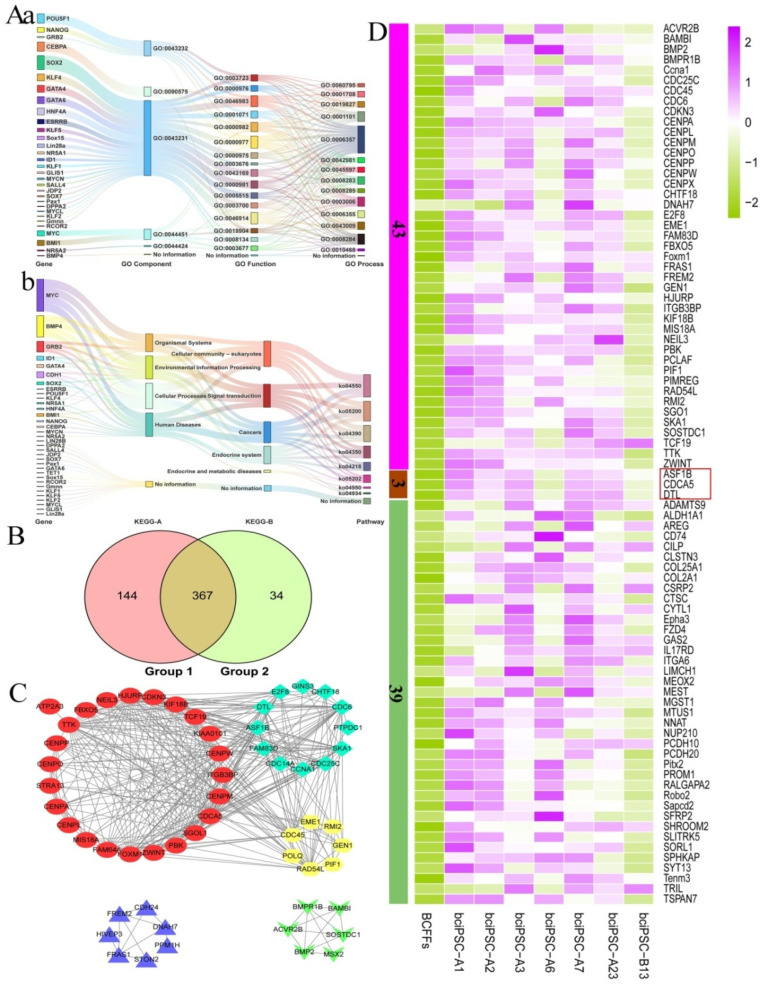
Mining of potentially pluripotency genes of bciPSCs. (**A**) (**a**) the attribution of the 34 induced genes in the GO database is displayed by an alluvial plot, (**b**) the attribution of the 34 induced genes in the KEGG database is displayed by a alluvial plot; (**B**) group 1 includes 511 genes; group 2 includes 401 genes; (**C**) the protein-protein interaction (PPI) network is composed of 56 genes; (**D**) a heat map of 85 genes. It is composed of 46 core genes and 42 supplementary genes, of which three genes are shared.

## Data Availability

The data that support the findings of this study are available from the corresponding author on reasonable request.

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
