# Peer review of "Establishment of Bactrian Camel Induced Pluripotent Stem Cells and Prediction of Their Unique Pluripotency Genes"

_ijms, 2023, doi:10.3390/ijms24031917_

Round 1
Reviewer 1 Report
Dear editor,
The article is focused on the generation and characterization of the Bactrian camel iPSCs.
General remarks:
1. English language editing is absolutely necessary as there are numerous grammatical mistakes. Also, reference format is occasionally wrong throughout the article.
2. Why retroviral vectors were used for iPSC generation? Lentiviral vectors are superior as some of them belong to the 3rd generation and they infect non-dividing cells.
3. It is unclear, why a clone in Figure 1Db is called “ES-like”? Pluripotent stem cells of different species and of different state (naïve or primed) look different, and there are no camel ES cells for comparison.
4. At L111-112 the authors claim that Nanog promoter is hypomethylated. The conclusion is based on just 6 sequences and the differences are not striking. It is necessary to expand bisulfite sequencing analysis to tens of strands and to apply statistical analysis. The current assumption is not based on data.
5. L127 What was the gender of the animal, XX or XY? If XX, X-chromosome inactivation status should be checked. Also, there should be metaphase plates and karyotype analysis: number of aneuploidy and tetraploid cells, at least 30 metaphase plates analyzed.
6. L178 Microarray data are, in my opinion, less precise, that transcriptome data. Camel iPSC transcriptome data should be compared with available transcriptomes not microarrays.
7. L295-297 The authors claim that they predicted pluripotency-related genes. It is impossible to make such predictions based on just two cell types – fibroblasts and iPSCs. Numerous cell types should be sequenced to be able to draw such a conclusion; it shall be removed and/or reformulated.
8. According to Figure 1F, the clones with least pluripotency marker expression (Nanog an Oct4) were selected for the pluripotency tests and transcriptome analysis. Why is that?
Overall, the data presented in the article is interesting. I’d recommend the article for publication after the major revision and English language proofreading.
Minor remarks:
L17 What is the meaning of “the most extensive host retroviruses”?
L25-26 “85 genes that could be used as potentially unique pluripotency genes to produce bciPSCs” – the genes uniquely expressed in iPSCs or that could be used for the iPSCs generation? If the latter, that statement is not supported by the article data.
L27-28 The sentence formulation is incorrect.
L43-44 There are many species with produced iPSCs. What was the criteria for the selection of references? I’d suggest to include all species or include only Artiodactyla.
L82-89 In Figure 1 Dd and De it seems that the bars are incorrect, as the size of the cells varies significantly. Figure 1 Вс – what is “F1”?
L94 What is a reporter gene? Is it just GFP that reports to the transduction efficiency, of it is under some pluripotency gene promoter?
L104-105 Both AP and Nanog expression test require cell death. How were they used for the removing differentiated cells from the alive cell culture?
L108-109 The sentence is not clear. Also, it is not clear, what is “relative expression”?
L127 What was the gender of the animal, XX or XY? How many metaphase plates analysed? How many aneuploidy and tetraploid cells?
L168-169 The sentence about the types of the diagrams is absolutely redundant.
L171 What is “intensive” distribution?
L269 The word “foreign” is incorrect here
L273 The demethylation of Nanog promoters was not proved!
L274 2n is not equal to 37
L281 What is the connection between gestation period and long teratoma formation?
L289 “a unique pluripotency-related gene” – just one gene?
L359 Clones, not “monoclones”
L367-369 “Based on the data” (?) three clones with practically no expression of Nanog and Oct4 were “randomly” (?) selected for further analyses. How is that possible? The other clones should have being selected!
L493-495 There is practically no application of camel in the human medicine, it should be removed.
Author Response
General remarks:
Point 1: English language editing is absolutely necessary as there are numerous grammatical mistakes. Also, reference format is occasionally wrong throughout the article.
Response 1: Thank you for your question. I edited the English language again and corrected the format of the references.
Point 2: Why retroviral vectors were used for iPSC generation? Lentiviral vectors are superior as some of them belong to the 3rd generation and they infect non-dividing cells.
Response 2: Thank you for your question. Retrovirus is the first viral vector used for gene transfer. It can infect a wide range of cells and various cell types of different biological species. It is suitable for the establishment of new species induce stem cells. It is also a virus vector owned by our laboratory. According to the actual needs of this study and the existing conditions in the laboratory, we did not try other types of virus vectors.
Point 3: It is unclear, why a clone in Figure 1Db is called “ES-like”? Pluripotent stem cells of different species and of different state (naïve or primed) look different, and there are no camel ES cells for comparison.
Response 3: Thank you for your question. I have modified "ES like clone" to "bciPSCs clone".
Point 4: At L111-112 the authors claim that Nanog promoter is hypomethylated. The conclusion is based on just 6 sequences and the differences are not striking. It is necessary to expand bisulfite sequencing analysis to tens of strands and to apply statistical analysis. The current assumption is not based on data.
Response 4: Thank you for your question. When designing methylation detection experiments, the number of CpG sequences detected in the literature consulted ranged from 6 to 23[1-6]. There is no CpG island in the NANOG promoter region of Bactrian Camel, so we only selected 6 CpG sequences that are relatively clustered in front of the Initiation codon for detection. The results showed that the degree of methylation of the NANOG promoter in bciPSCs became lower, which would not occur if they were not reprogrammed[3].The degree of demethylation in different regions of the gene promoter is different[1-6] and no quantifiable standard can be found, so it is defined as "highly unmethylated" according to the first four sequences at that time. Considering the actual situation and the preciseness of expression, we changed the original text of the article to " Bisulfite genomic sequencing analyses revealed that the NANOG promoter of the three bciPSC lines was demethylated to varying degrees (Figure 2A).".
References
[1] Cao H, Yang P, Pu Y, Sun X, Yin H, Zhang Y, Zhang Y, Li Y, Liu Y, Fang F, Zhang Z, Tao Y, Zhang X. Characterization of bovine induced pluripotent stem cells by lentiviral transduction of reprogramming factor fusion proteins. Int J Biol Sci. 2012;8(4):498-511.
[2] Liu H, Zhu F, Yong J, Zhang P, Hou P, Li H, Jiang W, Cai J, Liu M, Cui K, Qu X, Xiang T, Lu D, Chi X, Gao G, Ji W, Ding M, Deng H. Generation of induced pluripotent stem cells from adult rhesus monkey fibroblasts. Cell Stem Cell. 2008 Dec 4;3(6):587-90.
[3] Choi KH, Park JK, Son D, Hwang JY, Lee DK, Ka H, Park J, Lee CK. Reactivation of Endogenous Genes and Epigenetic Remodeling Are Barriers for Generating Transgene-Free Induced Pluripotent Stem Cells in Pig. PLoS One. 2016 Jun 23;11(6):e0158046.
[4] Honda A, Hirose M, Hatori M, Matoba S, Miyoshi H, Inoue K, Ogura A. Generation of induced pluripotent stem cells in rabbits: potential experimental models for human regenerative medicine. J Biol Chem. 2010 Oct 8;285(41):31362-9.
[5] Ren J, Pak Y, He L, Qian L, Gu Y, Li H, Rao L, Liao J, Cui C, Xu X, Zhou J, Ri H, Xiao L. Generation of hircine-induced pluripotent stem cells by somatic cell reprogramming. Cell Res. 2011 May;21(5):849-53.
[6] Yu J, Vodyanik MA, Smuga-Otto K, Antosiewicz-Bourget J, Frane JL, Tian S, Nie J, Jonsdottir GA, Ruotti V, Stewart R, Slukvin II, Thomson JA. Induced pluripotent stem cell lines derived from human somatic cells. Science. 2007 Dec 21;318(5858):1917-20.
Point 5: L127 What was the gender of the animal, XX or XY? If XX, X-chromosome inactivation status should be checked. Also, there should be metaphase plates and karyotype analysis: number of aneuploidy and tetraploid cells, at least 30 metaphase plates analyzed.
Response 5: Thank you for your question. Because there is no standard banding pattern map for bactrian camels, we only carried out chromosome number statistics for the relatively clear split phase. The literature shows that this method is also feasible[1,2]. The number of karyotype analysis in the literature related to the establishment of iPSCs ranges from 4[3,4]to 50 (generally 20) [1,5-7], and some of them are not even published[8,9]. The number of three bciPSCs for mitotic-phase spreads assessment was 22 (bciPSCs-A2), 25 (bciPSCs-A3), and 20 (bciPSCs-B13), respectively.Their karyotypes are normal. We did not carry out sex determination gene identification and G-banding karyotype analysis on the cells used, so we could not determine the chromosome category. To sum up, our karyotype analysis test can show that the bciPSCs obtained are diploid. Considering the preciseness of the article, we have included the specific number of statistics in the article.
References
[1] Liu J, Balehosur D, Murray B, Kelly JM, Sumer H, Verma PJ. Generation and characterization of reprogrammed sheep induced pluripotent stem cells. Theriogenology. 2012 Jan 15;77(2):338-46.e1.
[2] Esteban MA, Xu J, Yang J, Peng M, Qin D, Li W, Jiang Z, Chen J, Deng K, Zhong M, Cai J, Lai L, Pei D. Generation of induced pluripotent stem cell lines from Tibetan miniature pig. J Biol Chem. 2009 Jun 26;284(26):17634-40.
[3] Osteil P, Tapponnier Y, Markossian S, Godet M, Schmaltz-Panneau B, Jouneau L, Cabau C, Joly T, Blachère T, Gócza E, Bernat A, Yerle M, Acloque H, Hidot S, Bosze Z, Duranthon V, Savatier P, Afanassieff M. Induced pluripotent stem cells derived from rabbits exhibit some characteristics of naïve pluripotency. Biol Open. 2013 May 10;2(6):613-28.
[4] Domingo-Prim J, Riera M, Burés-Jelstrup A, Corcostegui B, Pomares E. Establishment of an induced pluripotent stem cell line (FRIMOi005-A) derived from a retinitis pigmentosa patient carrying a dominant mutation in RHO gene. Stem Cell Res. 2019 Jul;38:101468.
[5] Choi KH, Park JK, Son D, Hwang JY, Lee DK, Ka H, Park J, Lee CK. Reactivation of Endogenous Genes and Epigenetic Remodeling Are Barriers for Generating Transgene-Free Induced Pluripotent Stem Cells in Pig. PLoS One. 2016 Jun 23;11(6):e0158046.
[6] Liu J, Balehosur D, Murray B, Kelly JM, Sumer H, Verma PJ. Generation and characterization of reprogrammed sheep induced pluripotent stem cells. Theriogenology. 2012 Jan 15;77(2):338-46.e1.
[7] Honda A, Hirose M, Hatori M, Matoba S, Miyoshi H, Inoue K, Ogura A. Generation of induced pluripotent stem cells in rabbits: potential experimental models for human regenerative medicine. J Biol Chem. 2010 Oct 8;285(41):31362-9.
[8] Montserrat N, Bahima EG, Batlle L, Häfner S, Rodrigues AM, González F, Izpisúa Belmonte JC. Generation of pig iPS cells: a model for cell therapy. J Cardiovasc Transl Res. 2011 Apr;4(2):121-30.
[9] Ezashi T, Telugu BP, Alexenko AP, Sachdev S, Sinha S, Roberts RM. Derivation of induced pluripotent stem cells from pig somatic cells. Proc Natl Acad Sci U S A. 2009 Jul 7;106(27):10993-8.
Point 6: L178 Microarray data are, in my opinion, less precise, that transcriptome data. Camel iPSC transcriptome data should be compared with available transcriptomes not microarrays.
Response 6: Thank you for your question. Transcriptome data can indeed be more accurate, but different sequencing units and sequencing depths will also lead to different results. Transcriptome data is usually raw data, and its analysis requires very professional knowledge. Microarray data requirements are relatively low, and can meet the test requirements. After trying two kinds of data, we finally chose microarray data for analysis.
Point 7: L295-297 The authors claim that they predicted pluripotency-related genes. It is impossible to make such predictions based on just two cell types – fibroblasts and iPSCs. Numerous cell types should be sequenced to be able to draw such a conclusion; it shall be removed and/or reformulated.
Response 7: Thank you for your question. It is possible to narrow the screening scope infinitely by increasing the types of comparison cells. However, there is no standard for selecting what type of cells and how many types of cells. If the selection method is incorrect, it will also cause difficulties in subsequent data analysis and loss of candidate genes. With the in-depth study of iPSCs, the number of inducible genes is increasing. Therefore, we changed our thinking and referred to the reported inducible genes to predict the potential pluripotent inducible genes. At the same time, it was supplemented from two aspects: gene expression ranking and RNA seq data direct analysis (GO term and KEGG pathway). We believe that in the case of only two cell types[1,2], it is theoretically feasible to predict by the above method.We have reformulated the content of this part and added descriptions of relevant contents.
References
[1] Ezashi T, Telugu BP, Alexenko AP, Sachdev S, Sinha S, Roberts RM. Derivation of induced pluripotent stem cells from pig somatic cells. Proc Natl Acad Sci U S A. 2009 Jul 7;106(27):10993-8.
[2] Liao J, Cui C, Chen S, Ren J, Chen J, Gao Y, Li H, Jia N, Cheng L, Xiao H, Xiao L. Generation of induced pluripotent stem cell lines from adult rat cells. Cell Stem Cell. 2009 Jan 9;4(1):11-5.
Point 8: According to Figure 1F, the clones with least pluripotency marker expression (Nanog an Oct4) were selected for the pluripotency tests and transcriptome analysis. Why is that?
Response 8: Thank you for your question. Because they all express pluripotent marker genes, we randomly selected clones. The expression of pluripotent marker genes is not our screening criteria.
Minor remarks:
Point 1: L17 What is the meaning of “the most extensive host retroviruses”?
Response 1: Thank you for your question. We would like to express that retroviruses have a large range of hosts. We have reformulated the content of this part.
Point 2: L25-26 “85 genes that could be used as potentially unique pluripotency genes to produce bciPSCs” – the genes uniquely expressed in iPSCs or that could be used for the iPSCs generation? If the latter, that statement is not supported by the article data.
Response 2: Thank you for your question. Based on the reported inducible genes and transcriptome data, we predicted the potential specific inducible genes of Bactrian Camels. The induction ability of these genes has not been verified. We changed the original text of the article to " By mining RNA sequence data, 85 potential unique pluripotent genes of Bactrian camels were predicted, which could be used as candidate genes for the production of bciPSC in the future.".
Point 3: L27-28 The sentence formulation is incorrect.
Response 3: Thank you for your question. We changed the original text of the article to " In conclusion, we generated bciPSCs for the first time and obtained their transcriptome information, expanding the iPSC genetic information database and exploring the applicability of iPSCs in livestock."
Point 4: L43-44 There are many species with produced iPSCs. What was the criteria for the selection of references? I’d suggest to include all species or include only Artiodactyla.
Response 4: Thank you for your question. We mainly refer to the literature on the establishment of iPSCs in animals for the first time. We have reformulated the content of this part.
Point 5: L82-89 In Figure 1 Dd and De it seems that the bars are incorrect, as the size of the cells varies significantly. Figure 1 Вс – what is “F1”?
Response 5: Thank you for your question. I have modified " F1" to " P1". We checked and modified the scale in Figure 1D.
Point 6: L94 What is a reporter gene? Is it just GFP that reports to the transduction efficiency, of it is under some pluripotency gene promoter?
Response 6: Thank you for your question. It just reports the transduction efficiency.
Point 7: L104-105 Both AP and Nanog expression test require cell death. How were they used for the removing differentiated cells from the alive cell culture?
Response 7: Thank you for your question. Cell lines with negative AP and NANONG test results will terminate subsequent culture, not remove differentiated cells. We have reformulated the content of this part.
Point 8: L108-109 The sentence is not clear. Also, it is not clear, what is “relative expression”?
Response 8: Thank you for your question. We changed the original text of the article to “SOX2 and NANOG expression was detected in seven bciPSC cell lines (Figure 1F).”.But we did not find "relative expression".
Point 9: L127 What was the gender of the animal, XX or XY? How many metaphase plates analysed? How many aneuploidy and tetraploid cells?
Response 9: Thank you for your question. We have answered in point 5 of general remarks.
Point 10: L168-169 The sentence about the types of the diagrams is absolutely redundant.
Response 10: Thank you for your question. We have reformulated the content of this part.
Point 11: L171 What is “intensive” distribution?
Response 11: Thank you for your question. We changed the original text of the article to " The violin chart demonstrated that the sequencing data had only a few abnormal values and was highly reliable (Figure S2D).".
Point 12: L269 The word “foreign” is incorrect here
Response 12: Thank you for your question. I have modified " foreign genes " to " exogenous genes ".
Point 13: L273 The demethylation of Nanog promoters was not proved!
Response 13: Thank you for your question. We have answered in point 4 of general remarks.
Point 14: L274 2n is not equal to 37
Response 14: Thank you for your question. I have modified " 2n=37 " to " 2n=74 ".
Point 15: L281 What is the connection between gestation period and long teratoma formation?
Response 15: Thank you for your question.Teratoma experiment can test the ability of iPSCs to differentiate into three germ layers, which is very similar to the process of embryonic development. It has been reported that teratoma can be used to study embryonic development[1,2]. Bactrian camels and horses have a longer gestation period and teratoma formation time than other species, so we suspect that there is a relationship between the two.
References
[1] McDonald D, Wu Y, Dailamy A, Tat J, Parekh U, Zhao D, Hu M, Tipps A, Zhang K, Mali P. Defining the Teratoma as a Model for Multi-lineage Human Development. Cell. 2020 Nov 25;183(5):1402-1419.e18.
[2] Nagy K, Sung HK, Zhang P, Laflamme S, Vincent P, Agha-Mohammadi S, Woltjen K, Monetti C, Michael IP, Smith LC, Nagy A. Induced pluripotent stem cell lines derived from equine fibroblasts. Stem Cell Rev Rep. 2011 Sep;7(3):693-702.
Point 16: L289 “a unique pluripotency-related gene” – just one gene?
Response 16: Thank you for your question. I have modified " a unique pluripotency-related gene " to " unique pluripotency-related genes ".
Point 17: L359 Clones, not “monoclones”
Response 17: Thank you for your question. I have modified " monoclones " to " clones ".
Point 18: L367-369 “Based on the data” (?) three clones with practically no expression of Nanog and Oct4 were “randomly” (?) selected for further analyses. How is that possible? The other clones should have being selected!
Response 18: Thank you for your question. We have answered in point 8 of general remarks and reformulated the content of this part.
Point 19: L493-495 There is practically no application of camel in the human medicine, it should be removed.
Response 19: Thank you for your question. We changed the original text of the article to “Considering the development of research on disease treatment of Bactrian camels, this research is expected to contribute to the transformation of regenerative medicine in the future.”.

Reviewer 2 Report
In the manuscript titled “Establishment of Bactrian Camel Induced Pluripotent Stem Cells and Prediction of Their Unique Pluripotency Genes” authors report establishing iPSC lines from the Bactrian camel. Bacterial camel fetal fibroblasts were reprogrammed using ectopic expression of mouse transgenes (Oct4, Sox2, Klf4, and c-Myc) and extensive characterization of the generated iPSC clones was performed.
However, the data presented in the manuscript is not convincing that the authors were successful in establishing fully reprogrammed iPSCs from this species.
1. Cellular morphology in the images, presented in figure 1D, is not consistent with the cellular morphology of the embryonic or induced pluripotent stem cells.
2. Higher acid phosphatase (AP) expression is not a conclusive indicator of successful iPSC reprogramming.
3. Though it is not very clear in the manuscript how RNA seq data was analyzed and the gene expressions were quantified, and why authors performed comparative analysis using microarray data from other species than RNA seq data. But from the following statement, it is apparent that the generated iPSCs lacked the expression of core pluripotency genes Oct4, Sox2, and Nanog.
“Moreover, Sox2 and NANOG were not considerably expressed, Klf4 and c-Myc genes were down-regulated in bciPSCs (Figure 4E). This result coincided with that of pig iPSCs[7]. However, we found the expression of the replacement genes increased, such as GRB2 for Oct4[53]; RCOR2[54], and GMNN[53] for Sox2; L-MYC for c-Myc[13]; and L-myc’s binding protein gene MYCBP (Figure 4E).”
The author's interpretation of the replacement genes and the interpretation of the citations within this statement is not accurate.
4. The positive expression of core pluripotency genes by immunocytochemistry analysis shown in figure 2 is not quantitative and could be due to transgenes expression or low expression of pluripotency genes in partially reprogrammed cells or both. Including a positive control of mouse or human iPSCs/ESCs would help in clarifying these results.
Overall, before the validation (test for differentiation potential) and genomic integrity testing (karyotyping) author should establish that their generated iPSC clones robustly express core pluripotency genes.
Author Response
Point 1: Cellular morphology in the images, presented in figure 1D, is not consistent with the cellular morphology of the embryonic or induced pluripotent stem cells.
Response 1: Thank you for your question. The bciPSCs clone (Db) [1,2] and undifferentiated cells (Dc, Dd, and Df) [2,3] in Figure 1D have been compared with the pictures in the existing literature. It can be shown that the cells of bciPSCs obtained by us are consistent with the existing induced stem cells.
References
[1] Afanassieff M, Tapponnier Y, Savatier P. Generation of Induced Pluripotent Stem Cells in Rabbits. Methods Mol Biol. 2016;1357:149-72.
[2] Osteil P, Tapponnier Y, Markossian S, Godet M, Schmaltz-Panneau B, Jouneau L, Cabau C, Joly T, Blachère T, Gócza E, Bernat A, Yerle M, Acloque H, Hidot S, Bosze Z, Duranthon V, Savatier P, Afanassieff M. Induced pluripotent stem cells derived from rabbits exhibit some characteristics of naïve pluripotency. Biol Open. 2013 May 10;2(6):613-28.
[3] Honda A, Hirose M, Hatori M, Matoba S, Miyoshi H, Inoue K, Ogura A. Generation of induced pluripotent stem cells in rabbits: potential experimental models for human regenerative medicine. J Biol Chem. 2010 Oct 8;285(41):31362-9.
Point 2: Higher acid phosphatase (AP) expression is not a conclusive indicator of successful iPSC reprogramming.
Response 2: Thank you for your question. AP detection is only one of the indicators to identify iPSC pluripotency. Because it is easy to operate and observe, we took AP detection as one of the main indicators in the stage of exploring the cultivation scheme. At the later stage of the experiment, we comprehensively identified bciPSCs through cell morphology, expression of pluripotent genes and markers, chromosome number, transcriptome sequencing and differentiation potential.
Point 3: Though it is not very clear in the manuscript how RNA seq data was analyzed and the gene expressions were quantified, and why authors performed comparative analysis using microarray data from other species than RNA seq data. But from the following statement, it is apparent that the generated iPSCs lacked the expression of core pluripotency genesOCT4,SOX2, andNANOG.
“Moreover, Sox2 and NANOG were not considerably expressed, Klf4 and c-Myc genes were down-regulated in bciPSCs (Figure 4E). This result coincided with that of pig iPSCs[7]. However, we found the expression of the replacement genes increased, such as GRB2 for Oct4[53]; RCOR2[54], and GMNN[53] for Sox2; L-MYC for c-Myc[13]; and L-myc’s binding protein gene MYCBP (Figure 4E).”
The author's interpretation of the replacement genes and the interpretation of the citations within this statement is not accurate.
Response 3: Thank you for your question. RNA seq and microarray data were analyzed using edger and GEO2R software respectively. RNA seq data visualization is the online software of sequencing company. We added this part to the article. Transcriptome data is usually raw data, and its analysis requires very professional knowledge. Microarray data requirements are relatively low, and can satisfy the test requirements. After trying two kinds of data, we finally chose microarray data for analysis.
In the detection of bciPSCs core pluripotent genes, we can only design detection primers for NANOG and Sox2 core pluripotent genes. It is also possible to detect two genes according to literature review[1]. According to the results of qRT-PCR(Figure 1F), the samples expressing core genes were screened for transcriptome sequencing. The identified transcriptome data are available, but the analysis results of the two core genes are not ideal. Considering the integrity of bactrian camel reference genome; the results of qRT PCR and RNA seq are 15.1% - 19.4% inconsistent[2,3]; the same results were also found in the establishment of pig iPSCs[4]. Combined with immunofluorescence and methylation detection experiments(Figure 2A,D), we believe that the core gene in bciPSCs is expressed, but it has not been effectively detected in transcriptome data due to unknown reasons. Therefore, we analyzed the expression of other pluripotent genes and genes that can replace classical inducers to perform induction function in RNA seq data, so as to prove the pluripotency of bciPSCs.
References
[1] Esteban MA, Xu J, Yang J, Peng M, Qin D, Li W, Jiang Z, Chen J, Deng K, Zhong M, Cai J, Lai L, Pei D. Generation of induced pluripotent stem cell lines from Tibetan miniature pig. J Biol Chem. 2009 Jun 26;284(26):17634-40.
[2] Everaert C, Luypaert M, Maag JLV, Cheng QX, Dinger ME, Hellemans J, Mestdagh P. Benchmarking of RNA-sequencing analysis workflows using whole-transcriptome RT-qPCR expression data. Sci Rep. 2017 May 8;7(1):1559.
[3] Teng M, Love MI, Davis CA, Djebali S, Dobin A, Graveley BR, Li S, Mason CE, Olson S, Pervouchine D, Sloan CA, Wei X, Zhan L, Irizarry RA. A benchmark for RNA-seq quantification pipelines. Genome Biol. 2016 Apr 23;17:74.
[4] Ezashi T, Telugu BP, Alexenko AP, Sachdev S, Sinha S, Roberts RM. Derivation of induced pluripotent stem cells from pig somatic cells. Proc Natl Acad Sci U S A. 2009 Jul 7;106(27):10993-8.
Point 4: The positive expression of core pluripotency genes by immunocytochemistry analysis shown in figure 2 is not quantitative and could be due to transgenes expression or low expression of pluripotency genes in partially reprogrammed cells or both. Including a positive control of mouse or human iPSCs/ESCs would help in clarifying these results.
Response 4: Thank you for your question. We detected the core pluripotent gene by qRT-PCR and the NANOG promoter methylation. As one of the identification indicators, immunocytochemistry analysis needs to verify the pluripotency of bciPSCs together with other identification indicators. A large number of documents show that the experimental design in this study is feasible[1-9].
References
[1] Han X, Han J, Ding F, et al. Generation of induced pluripotent stem cells from bovine embryonic fibroblast cells [J]. Cell Research, 2011 Oct, 21(10): 1509-12.
[2] Liao J, Cui C, Chen S, et al. Generation of induced pluripotent stem cell lines from adult rat cells [J]. Cell Stem Cell, 2009 Jan 9, 4(1): 11-5.
[3] Takahashi K, Yamanaka S. Induction of Pluripotent Stem Cells from Mouse Embryonic and Adult Fibroblast Cultures by Defined Factors [J]. Cell, 2006 Aug 25, 126(4): 663-76.
[4] Yu J, Vodyanik M, Smuga-Otto K, et al. Induced pluripotent stem cell lines derived from human somatic cells [J]. Science, 2007 Dec 21, 318(5858): 1917-20.
[5] Ezashi T, Telugu B, Alexenko A, et al. Derivation of induced pluripotent stem cells from pig somatic cells [J]. Proc Natl Acad Sci U S A, 2009 Jul 7, 106(27): 10993-8.
[6] Nagy K, Sung H, Zhang P, et al. Induced Pluripotent Stem Cell Lines Derived from Equine Fibroblasts [J]. Stem Cell Reviews, 2011 Sep, 7(3): 693-702.
[7] Liu J, Balehosur D, Murray B, et al. Generation and characterization of reprogrammed sheep induced pluripotent stem cells [J]. Theriogenology, 2012 Jan 15, 77(2): 338-46.
[8] Liu H, Zhu F, Yong J, et al. Generation of Induced Pluripotent Stem Cells from Adult Rhesus Monkey Fibroblasts [J]. Cell Stem Cell, 2008 Dec 4, 3(6): 587-90.
[9] Buehr M, Meek S, Blair K, et al. Capture of authentic embryonic stem cells from rat blastocysts [J]. Cell, 2008 Dec 26, 135(7): 1287-98.

Round 2
Reviewer 1 Report
Dear editor,
I disagree with the number of Nanog CpG methylation sequences, 6 is to low to draw any meaningful conclusions, even the one in the corrected MS: "demethylated to varying degrees". The author's argument about the literature data is not applicable - if someone used just several sequences instead of tens, it's just bad data.
The other problem that was not addressed - the choice of iPSC clones for the analysis. The ones with the highest pluripotency marker expression should have boon used.
Overall the data on camel iPSCs generation is interesting and scientifically sound. I can recommend the MS for publication.
Author Response
Point 1: I disagree with the number of Nanog CpG methylation sequences, 6 is to low to draw any meaningful conclusions, even the one in the corrected MS: "demethylated to varying degrees". The author's argument about the literature data is not applicable - if someone used just several sequences instead of tens, it's just bad data.
Response 1: Thank you for your question. As the laboratory of the university is under closed management of the epidemic situation, the methylation experiment cannot be further improved at present. In view of the strict requirements of experts on this verification method, we will delete this part of the experimental content to avoid disputes.
Point 2: The other problem that was not addressed - the choice of iPSC clones for the analysis. The ones with the highest pluripotency marker expression should have boon used.
Response 2: Thank you for your question. We reiterate the reason again, because they all express pluripotent marker genes, so we randomly selected clones. The expression of pluripotent marker genes is not our screening criteria.
Assuming that the selection is based on the results of qPCR, will it be more helpful to prove that we have successfully established bciPSCs by verifying the three clones with the lowest pluripotent gene expression?
However, our clones were randomly selected and did not use the results of qPCR as the basis for selection. The expression of pluripotent genes in the three clones was lower than that in the other clones, which was purely coincidental.

Reviewer 2 Report
I am not convinced by the author's response to my previous serious concerns. In my experience, the cellular morphology presented in figure 1D panels is not consistent with fully reprogrammed iPSCs. I have compared the images in figure 1D with the published iPSC images the authors have referenced in their response. I don’t see that they are similar.
The qPCR-based relative quantities of the “Endo Sox2” and “Endo Nanog” are significantly inconsistent within the iPSC clones tested. Why did the clones “bciPSC-A6” and bciPSC-A23”, which showed the highest relative qualities of the aforementioned Sox2 and Nanog not followed in the Nanog methylation assay? Also, why Oct4 expression was not quantified by qPCR?
Another major concern is that the core pluripotency genes Oct4, Sox2, and Nanog were not found expressed in their RNA seq data. If authors do not have sufficient expertise in RNA seq analysis, they should take the help of an expert to clearly understand the RNA seq data.
Overall, I am not convinced that the manuscript is acceptable for publication in its current form.
Author Response
Point 1: I am not convinced by the author's response to my previous serious concerns. In my experience, the cellular morphology presented in figure 1D panels is not consistent with fully reprogrammed iPSCs. I have compared the images in figure 1D with the published iPSC images the authors have referenced in their response. I don’t see that they are similar.
Response 1: Thank you for your question. Thank you for your question. For Figure 1Da, please refer to Figure 3b in Reference 1[1], Figure 1B and Figure 2Ae in Reference 2[2], Figure 1E and Figure 2Ae in Reference 5[5]; For Figure 1Db, please refer to Figure 3a in Reference 1[1], Figure 1B and Figure 2A (b, c) in Reference 2[2], and Figure 2C in Reference 3 (17 days) [3]; For Figure 1Dc, please refer to Figure 1C and Figure 2A in Reference 4[4]. Figure 1 Dd and Df were taken after 12h(with trophoblast) of passage by recombinant enzyme digestion , and no comparative photos of this cell density were found in the literature. So we will delete the corresponding photos.
References
[1] Afanassieff M, Tapponnier Y, Savatier P. Generation of Induced Pluripotent Stem Cells in Rabbits. Methods Mol Biol. 2016;1357:149-72.
[2] Osteil P, Tapponnier Y, Markossian S, Godet M, Schmaltz-Panneau B, Jouneau L, Cabau C, Joly T, Blachère T, Gócza E, Bernat A, Yerle M, Acloque H, Hidot S, Bosze Z, Duranthon V, Savatier P, Afanassieff M. Induced pluripotent stem cells derived from rabbits exhibit some characteristics of naïve pluripotency. Biol Open. 2013 May 10;2(6):613-28.
[3] Huang P, Zhu J, Liu Y, Liu G, Zhang R, Li D, Pei D, Zhu P. Identification of New Transcription Factors that Can Promote Pluripotent Reprogramming. Stem Cell Rev Rep. 2021 Dec;17(6):2223-2234. doi: 10.1007/s12015-021-10220-z. Epub 2021 Aug 26.
[4] Honda A, Hirose M, Hatori M, Matoba S, Miyoshi H, Inoue K, Ogura A. Generation of induced pluripotent stem cells in rabbits: potential experimental models for human regenerative medicine. J Biol Chem. 2010 Oct 8;285(41):31362-9.
[5] Liu M, Zhao L, Wang Z, Su H, Wang T, Yang G, Chen L, Wu B, Zhao G, Guo J, Yang Z, Zhang J, Hao C, Ma T, Song Y, Bao S, Zuo Y, Li X, Cao G. Generation of Sheep Induced Pluripotent Stem Cells With Defined DOX-Inducible Transcription Factors via piggyBac Transposition. Front Cell Dev Biol. 2021 Dec 16;9:785055.
Point 2: The qPCR-based relative quantities of the “Endo Sox2” and “Endo Nanog” are significantly inconsistent within the iPSC clones tested. Why did the clones “bciPSC-A6” and bciPSC-A23”, which showed the highest relative qualities of the aforementioned Sox2 and Nanog not followed in the Nanog methylation assay? Also, why Oct4 expression was not quantified by qPCR?
Response 1: Thank you for your question. Because the reprogramming degree of seven bciPSCs was different, the expression amount of two genes was different.
After detecting the pluripotent genes of seven bciPSCs by qPCR, we randomly selected three clones for subsequent culture and identification. Therefore, there is no test result of cloning "bciPSC-A6" and "bciPSC-A23".As the laboratory of the university is under closed management of the epidemic situation, the methylation experiment cannot be further improved at present. In view of the strict requirements of experts on this verification method, we will delete this part of the experimental content to avoid disputes.
After trying, we could not design a suitable Oct4 detection primer (the band is not single), so we did not detect the gene.
Point 3: Another major concern is that the core pluripotency genes Oct4, Sox2, and Nanog were not found expressed in their RNA seq data. If authors do not have sufficient expertise in RNA seq analysis, they should take the help of an expert to clearly understand the RNA seq data.
Response 1: Thank you for your question. The pluripotent gene was detected by qPCR before the sample was sent for sequencing, and the result of immunofluorescence of the pluripotent gene was positive. This situation has also occurred in published articles, in which we also explained the possible reasons. The verification of bciPSCs' pluripotency should be mutually verified by morphology, pluripotent gene, pluripotent marker protein, AP, telomerase gene detection, karyotype analysis, embryoid differentiation, teratoma test and transcriptome sequencing. Deficiencies of bciPSCs in one test can be verified by other tests. Transcriptome data proved bciPSCs from the overall level, pluripotency related pathways and gene expression, DNA methyltransferase, DNA methyl oxidase, histone demethylase and histone genes. Based on all the verification results, we believe that the conclusions can be supported.
